# Use of volatile agents for sedation in the intensive care unit: A national survey in France

**Raiko Blondonnet**[1,2]*, **Audrey Quinson**[1], **Céline Lambert**[3], **Jules Audard**[1,2], **Thomas Godet**[1], **Ruoyang Zhai**[2], **Bruno Pereira**[3], **Emmanuel Futier**[1,2], **Jean-Etienne Bazin**[1], **Jean-Michel Constantin**[4], **Matthieu Jabaudon**[1,2,5]

**1** Department of Perioperative Medicine, CHU Clermont-Ferrand, Clermont-Ferrand, France, **2** GReD, CNRS, INSERM, Université Clermont Auvergne, Clermont-Ferrand, France, **3** Biostatistical and Data Management Unit, Department of Clinical Research and Innovation, CHU Clermont-Ferrand, Clermont-Ferrand, France, **4** Department of Anesthesiology and Critical Care, Sorbonne University, GRC 29, AP-HP, DMU DREAM, Pitié-Salpêtrière Hospital, Paris, France, **5** Division of Allergy, Pulmonary, and Critical Care Medicine, Department of Medicine, Vanderbilt University Medical Center, Nashville, Tennessee, United States of America

\* rblondonnet@chu-clermontferrand.fr

## Abstract

### Background

Current intensive care unit (ICU) sedation guidelines recommend strategies using non-benzodiazepine sedatives. This survey was undertaken to explore inhaled ICU sedation practice in France.

### Methods

In this national survey, medical directors of French adult ICUs were contacted by phone or email between July and August 2019. ICU medical directors were questioned about the characteristics of their department, their knowledge on inhaled sedation, and practical aspects of inhaled sedation use in their department.

### Results

Among the 374 ICUs contacted, 187 provided responses (50%). Most ICU directors (73%) knew about the use of inhaled ICU sedation and 21% used inhaled sedation in their unit, mostly with the *Anaesthetic Conserving Device* (*AnaConDa*, Sedana Medical). Most respondents had used volatile agents for sedation for <5 years (63%) and in <20 patients per year (75%), with their main indications being: failure of intravenous sedation, severe asthma or bronchial obstruction, and acute respiratory distress syndrome. Sevoflurane and isoflurane were mainly used (88% and 20%, respectively). The main reasons for not using inhaled ICU sedation were: "device not available" (40%), "lack of medical interest" (37%), "lack of familiarity or knowledge about the technique" (35%) and "elevated cost" (21%). Most respondents (80%) were overall satisfied with the use of inhaled sedation. Almost 75% stated that inhaled sedation was a seducing alternative to intravenous sedation.

**Data Availability Statement:** All relevant data are within the paper and its Supporting Information files.

**Funding:** This work was supported by internal funding of the Department of Perioperative medicine, CHU Clermont-Ferrand, France.

**Competing interests:** The authors have read the journal's policy and the authors of this manuscript have the following competing interests: MJ is a principal investigator of the SEvoflurane for Sedation in ARds (SESAR) (ClinicalTrials.gov Identifier: NCT04235608) and the ISCA study (ClinicalTrials.gov Identifier: NCT04383730), which are co-funded and funded, respectively, by grants from Sedana Medical. JMC and MJ received fees from Sedana Medical for participation in a scientific advisory panel in 2019; MJ received consulting fees from Abbvie in 2020. Neither Sedana Medical or Abbvie has no influence in the study and collection, analysis, and interpretation of data and in writing of the current study. There are no patents, products in development or marketed products to declare.This does not alter our adherence to PLOS ONE policies on sharing data and materials.

**Abbreviations:** ARDS, Acute respiratory distress syndrome; COVID-19, 2019 Coronavirus Disease; ICU, Intensive care unit; STROBE, Strengthening the reporting of observational studies in epidemiology.

## Conclusion

This survey highlights the widespread knowledge about inhaled ICU sedation in France but shows its limited use to date. Differences in education and knowledge, as well as the recent and relatively scarce literature on the use of volatile agents in the ICU, might explain the diverse practices that were observed. The low rate of mild adverse effects, as perceived by respondents, and the users' satisfaction, are promising for this potentially important tool for ICU sedation.

## Background

In the intensive care unit (ICU), sedation is used to improve comfort and tolerance during mechanical ventilation, invasive diagnostic and therapeutic interventions or nursing care [1]. Recent surveys of sedation practices report that midazolam and propofol remain mostly used for ICU sedation [2]. However, the literature highlights numerous adverse effects of such intravenous sedatives, including prolonged mechanical ventilation and ICU duration of stay, increased delirium, and risk of hemodynamic failure or severe syndromes such as propofol infusion syndrome [3–7]. The ideal sedative agent should act rapidly, provide effective sedation without side effects, not accumulate and allow fast awakening when interrupted. Current international guidelines recommend that sedation strategies using non-benzodiazepine drugs should be preferred over sedation with benzodiazepines to improve clinical outcomes in mechanically ventilated adult patients [8]. In some national guidelines, such as in Germany, the use of volatile anesthetics is also considered a feasible option [9]. Since the development of anesthetic reflectors, such as the *Anaesthetic Conserving Device* (AnaConDa, Sedana Medical, Danderyd, Sweden) and the *Mirus* (Carelide GmbH, Mouvaux, France), inhaled sedation has become more popular in the ICU [10–12]. These devices are suitable with modern critical care ventilators without extraordinary expense or technical difficulty, while ensuring the safety of both patients and ICU care providers and the protection of the workplace environment [13–16]. Volatile anesthetics have a bronchodilatory effect [17, 18] and may be protective for some organs, such as the heart [19, 20] and the kidney [21–23]. Moreover, animal and human studies support the protective effects of a pretreatment with volatile anesthetics before prolonged ischemia of the liver [24], the brain [25] or the heart [19]. Preclinical studies have also shown that inhaled sevoflurane could improve gas exchange [26, 27], reduce alveolar oedema [26, 27], and attenuate pulmonary and systemic inflammation [28] in experimental models of acute respiratory distress syndrome (ARDS). In a pilot randomised controlled trial, early use of inhaled sevoflurane was associated with improved oxygenation and a reduction of some proinflammatory markers and of a marker of lung epithelial injury in patients with ARDS, compared to intravenous midazolam [29].

Although volatile anesthetics could be part of the modern management of sedation of critically ill patients, data on their use in the ICU setting are scarce. This survey was undertaken to explore the use of volatile agents for ICU sedation in France, to assess the indications and modalities of their use by ICU care providers, and to understand the potential reasons against their use, if any.

## Methods

### Survey development

This investigator-initiated survey did not require approval by an Ethics Committee, as per French law. A 26-item questionnaire was developed with questions designed by the authors

(RB, AQ, JEB, JMC, MJ) (Supplemental Content 1 of S1 File); it covered three categories: general characteristics of the ICU; general data on inhaled sedation use, and practical aspects of inhaled ICU sedation.

## Survey sample

All French ICUs were identified and contacted unless they were pediatric ICUs. The survey was conducted between July and August 2019. Almost all responses were recorded orally. After short information about the survey design and objectives, the medical director of each ICU was exclusively questioned. Completion of the survey took approximately five minutes. After a first phone contact, and upon request of the ICU director, some questionnaires were emailed individually. In the absence of a reply after one month, a second contact with the ICU director was performed by telephone and followed by a third call when necessary.

## Statistical analysis

Statistical analysis was performed in compliance with the Strengthening the Reporting of Observational Studies in Epidemiology (STROBE) checklist [30]. Results were requested both in a descriptive manner and, for all closed questions, in absolute numbers or percentages. Responses to each questionnaire were recorded into a Microsoft Excel database and analysed using Stata software version 15 (StataCorp, College Station, Texas, USA). Categorical parameters were expressed as frequencies and associated percentages, and continuous data as mean ± standard deviation or median [interquartile range], according to statistical distribution. The Gaussian distribution was verified by the Shapiro-Wilk test.

## Results

Out of the 405 ICUs located in France [31], 31 pediatric ICUs were excluded and 374 adult ICU directors were directly questioned. A total of 50% (187/405) of the questionnaires were recorded; 81% (152/187) orally and 19% (35/187) electronically. There were 24% (45/187) of answers from teaching hospitals, 60% (111/187) from general hospitals, 14% (27/187) from private medical centers, and 2% (4/187) from military hospitals. Among participating ICUs, 91% (170/187) were mixed (medical and surgical) ICUs, 8% (15/187) were medical ICUs and 1% (2/187) were burn centers. General characteristics of ICU respondents are reported in Supplemental Contents 2 and 3 of S1 File.

Among the respondents, 73% (137/187) declared they knew about the use of inhaled sedation in the ICU. Of these, 98% (134/137) knew the *AnaConDa* device and 15% (20/137) knew the *Mirus* device. Twenty-one percent (40/187) of respondents declared they routinely performed inhaled sedation in their unit, 90% (36/40) with the *AnaConDa* and 10% (4/40) with the *Mirus*. However, no respondent had both devices available. Sixty-three percent declared they had used volatile agents for sedation for less than five years. In 45% (18/40) of respondents, inhaled sedation was performed only by a few practitioners from the medical team. Seventy-five percent (30/40) of the respondents declared they used inhaled ICU sedation in less than 20 patients per year (Fig 1).

In case neither the *AnaConDa* nor the *Mirus* were available in their hospital, 23% (34/147) of respondents declared they had already borrowed an anesthesia machine from the operating room to deliver inhaled sedation with volatile anesthetics in an ICU patient. The three main reasons for not using inhaled ICU sedation were: "device not available" (40% (58/147)), "no clear clinical benefit" (37% (55/147)), and "lack of familiarity or knowledge about the technique" (35% (51/147)). Seven percent of respondents (10/147) mentioned the potential risk of halogenated-induced atmospheric pollution, and only one respondent answered that he did

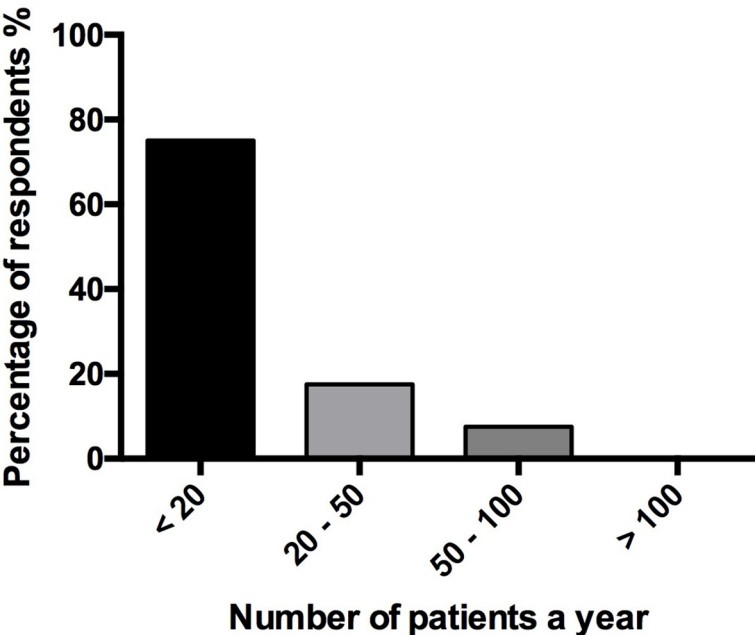

**Fig 1. Number of patients in whom inhaled sedation was used each year in responding intensive care units in which a dedicated device was available.** Data are represented in %.

not use volatile agents because of their potential adverse effects. Other indications evoked by respondents for not using inhaled ICU sedation are summarised in Fig 2.

The three main indications reported by the respondents were: failure of intravenous sedation (75% (30/40)), severe asthma (75% (30/40)), and ARDS (65% (26/40)). Other indications evoked by respondents for using inhaled ICU sedation are summarised in Fig 3. Sixty-nine percent (129/187) of respondents said they were aware of potential benefits of inhaled ICU

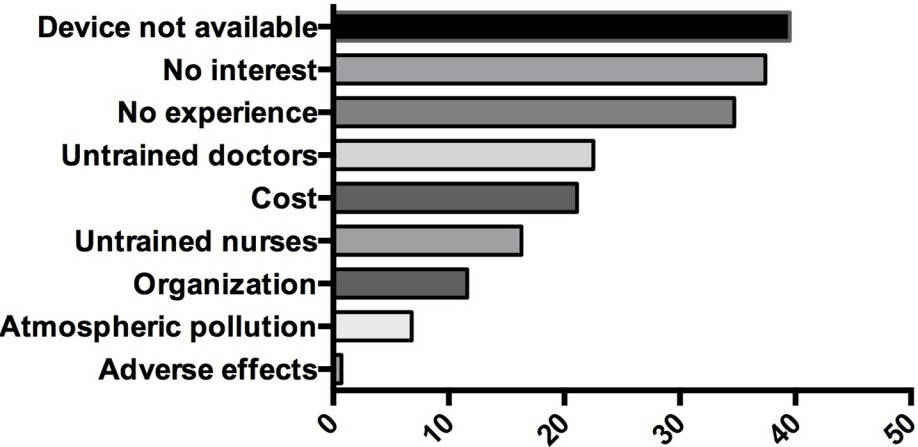

**Fig 2. Reasons reported for not using inhaled sedation in surveyed intensive care units.** (n = 147) *Data are represented in %.*

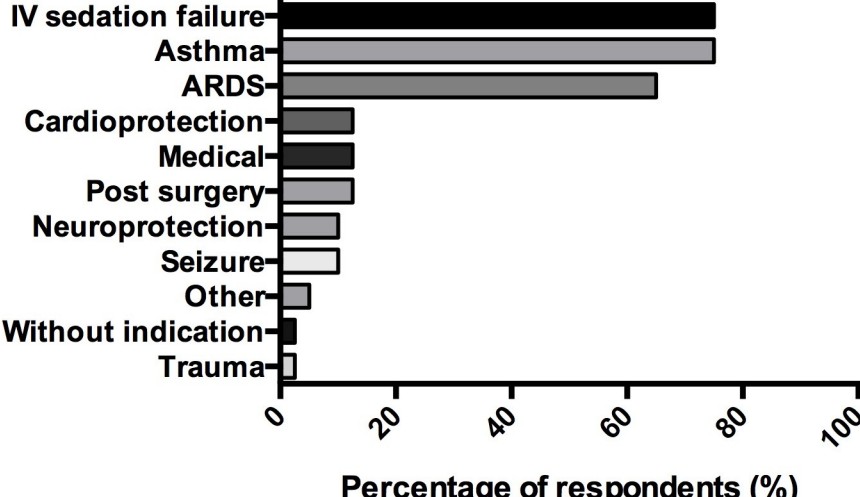

**Fig 3. Indications of volatile anesthetics for intensive care sedation, as reported by users.** (n = 40) *Data are represented in %. ARDS: Acute Respiratory Distress Syndrome; IV: Intravenous.*

sedation as reported in Fig 4. The main benefit being reported was bronchodilation (88% (114/129)), both by respondents who had a device suitable for inhaled ICU sedation available in their unit (90%, 35/39) and those who did not (88%, 79/90).

Eighty-four percent (157/187) of respondents declared they had a written protocol for sedation in their institution, whereas only 43% (17/40) of units in which volatile agents were used had a specific protocol for inhaled sedation. One-fifth (39/187) of respondents declared they had received specific training on inhaled sedation; 69% (27/39) and 64% (25/39) of this training originated from the companies developing the *AnaConda* or the *Mirus* devices and through scientific conferences, respectively.

Sevoflurane or isoflurane were mainly used by the respondents (88% (35/40) or 20% (8/40), respectively), and three respondents answered they used both. Desflurane was not used by any

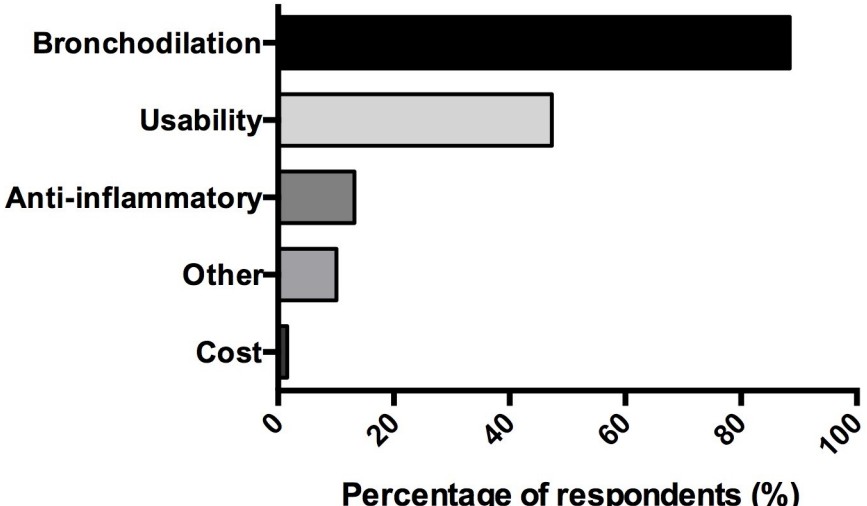

**Fig 4. Properties of volatile anesthetics for inhaled intensive care unit sedation, as reported by respondents.** (n = 129) *Data are represented in %.*

respondent. In 93% (37/40) of ICUs, respondents reported they usually combined opioid-based analgesia with inhaled sedation (73%, (29/40) with sufentanil, 40% (16/40) with remifentanil, and 3% (1/40) with fentanyl). Almost half (48% (19/40)) of respondents answered that they would combine inhaled sedation with continuous intravenous administration of at least one other hypnotic agent (38% (15/40) propofol, 20% (8/40) midazolam, 8% (3/40) dexmedetomidine, and 3% (1/40) ketamine).

Inhaled sedation was used along with controlled ventilation modes in all ICUs and 53% (21/40) of respondents using inhaled sedation reported they would also use inhaled sedation during pressure support ventilation in intubated patients. One respondent declared it has already used inhaled sedation during noninvasive ventilation. Most respondents (79%, (31/40)) answered they would use validated scales or scores, rather than end-tidal gas concentration monitoring, to titrate their sedation goals. The Richmond Agitation Sedation Scale (RASS) was used in 90% (36/40) of ICUs; 35% (14/40) of respondents declared they also monitored the bispectral index (BIS-ASPECT-A-2000; Aspect Medical Systems, Norwood, USA), and 25% (10/40) of ICUs monitored end-tidal gas concentration to titrate sedation. No ICU director reported they measured plasma concentrations of volatile anesthetics or their metabolites when monitoring inhaled sedation.

Sixty-three percent (25/40) of respondents answered they did not specifically set any maximal duration for inhaled sedation in their ICU patients. Thirty-three percent (13/40) of respondents declared they would interrupt inhaled sedation when patients are entering the process of weaning from ventilation. Five percent (2/40) of respondents declared they would systematically stop inhaled sedation within a maximum of five days.

Absolute contraindications for inhaled sedation could be reported by 53% (99/187) of respondents (Fig 5). At least one adverse effect attributable to volatile anesthetics already occurred in 28% (11/40) surveyed ICUs. Malignant hyperthermia, hypercapnic acidosis, diabetes insipidus, renal failure, cholestasis, arrhythmia, hemodynamic failure, and dysnatremia were reported as potential adverse effects. Five percent (2/40) of respondents mentioned that some nurses may have developed headaches after they had used volatile agents in their units.

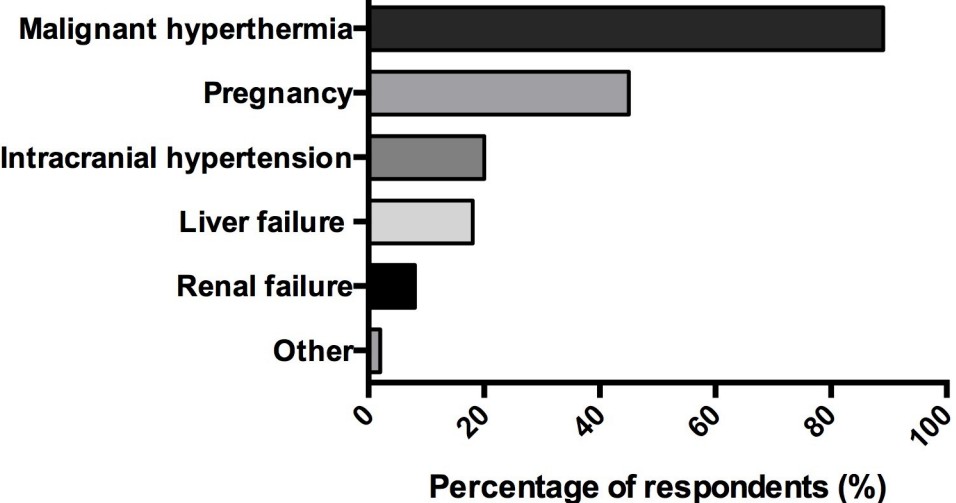

**Fig 5. Contraindication of volatile anesthetics for intensive care sedation, as reported by users.** (n = 99) *Data are represented in %.* Overall satisfaction with the use of inhaled sedation among the users is represented in Fig 6.

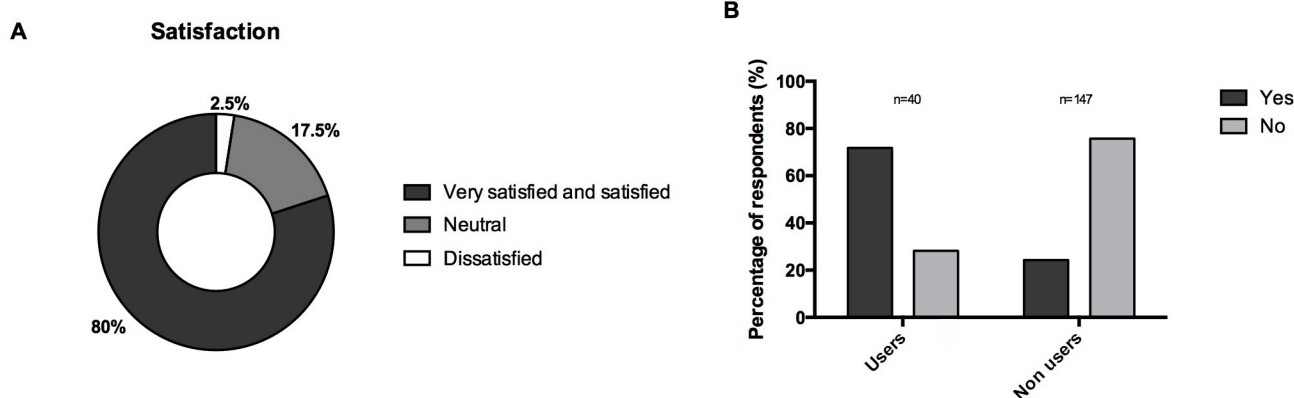

**Fig 6.** (A) Overall satisfaction of respondents who had a device suitable for inhaled sedation available in their intensive care unit regarding the use of inhaled sedation. (n = 40) (B) Answers to the question « Do you think that inhaled sedation is an interesting alternative to intravenous sedation in the intensive care unit? » by users and non-users of the technique. (n = 187) *Data are represented in %.*

## Discussion

This survey is the first to investigate the use of inhaled sedation and to assess the spread of knowledge about the technique in French ICUs. The high response rate (50%) to the survey suggests this was a relatively representative sample, especially because the distribution of hospital types within respondents was broadly comparable to the distribution of hospitals in France [32].

Differences in practices between medical and polyvalent ICUs, as well as disparities in knowledge on inhaled sedation between responders, could be explained by the various proportions of anesthesiologists-intensivists working in these units. In France, a combined anesthesiology, critical care and perioperative medicine specialty track trains "anesthesiologists-intensivists" who can subsequently work in the ICU, in the operative room, or in both. A distinct specialty track trains French "medical intensivists" who can work in ICUs but are not anesthesiologists. Anesthesiologists-intensivists are more likely to be trained to use volatile anesthetics during their residency program that involves more time working in the operating room. Unfortunately, the proportion of ~~staff~~ anesthesiologists-intensivists in each participating unit was not assessed in this survey. Differences in education and knowledge might also explain why many respondents reported poor medical interest (37%) for inhaled sedation use and lack of familiarity (35%) with the technique as main reasons for not using inhaled sedation in their units. Even if the recent and relatively scarce literature on volatile agents used in the ICU could explain these answers, this survey highlights a real interest in, and potentially a new application for, inhaled sedation in multiple French ICUs. However, the interest developed by the physicians may be impacted by their working environment. Indeed, respondents from general hospitals declared they were less interested by inhaled sedation than their colleagues from university hospitals. This could be explained, at least in part, because university hospitals, including major academic medical centers, may offer more advanced clinical capabilities and opportunities for medical research, education, and innovation [33].

One fifth of intensivists answered that they did not use inhaled sedation because of its related cost. Even if inhaled ICU sedation might sometimes be perceived as more costly [34, 35], reductions in indirect costs through shortened times to liberation from mechanical ventilation, patient recovery, and ICU discharge, as well as reduced needs for sedative and opioid agents, are plausible but often difficult to demonstrate [13, 34–37]. Regarding direct costs, in

some patients in whom sedation is deemed more difficult, inhaled sedation may significantly decrease costs of sedation, with daily costs being similar when using midazolam or isoflurane, and the cost difference could favor inhaled sedation in the subgroup of patients who required high doses of midazolam [34].

Main indications and advantages reported through this survey are in accordance with previously published evidence [15]. Most users are interested by the rapid onset of volatile anesthetics, their low metabolism and blood solubility, while avoiding tolerance or addiction phenomena and allowing fast awakening, as compared to current intravenous sedation practices. Indeed, in a recent meta-analysis by Jerath et al., time to extubation was reduced with volatile-based sedation, compared to intravenous sedation with either midazolam or propofol [38].

Half of physicians declared they used volatile agents in spontaneously breathing patients under pressure support ventilation. Nevertheless, adding the *AnaConDa* itself to the respiratory circuit may increase work-of-breathing, due to a dead space effect that is only partially explained by its larger internal volume. However, simultaneous sevoflurane inhalation decreases this increase in work-of-breathing, suggesting specific effects of sevoflurane through decreased carbon dioxide reflection [39] rather than through sevoflurane induced-bronchodilation, as previously suggested [40]. There is indeed growing evidence supporting that light sedation with sevoflurane through the *AnaConDa* is feasible in severely ill ICU patients [38].

Most surveyed ICU directors reported they did not set any maximal duration for inhaled ICU sedation in their patients. Mesnil et al. demonstrated that sevoflurane delivered through a dedicated device was a safe and effective alternative to intravenous sedation in ICU patients receiving under sedation for a median time of 50 [39–71] hours [13]. In addition, in this study, inhaled sevoflurane was associated with decreased awakening and extubation times, postextubation morphine consumption, and increased awakening quality [13]. Similar results were found with isoflurane [36]. Recent research suggests that volatile anesthetics could even have some protective effects on the kidney by attenuating renal tubular necrosis and decreasing the nephrotoxic effects of proinflammatory cytokines [21, 23]. A recent work that studied longer exposures has reported cases of nephrogenic diabetes insipidus (NDI) in patients under inhaled sedation with sevoflurane, which could be related to high doses and long durations of use [41]. In this cohort, patients who developed NDI were exposed for longer durations and with higher end-tidal concentrations of sevoflurane than those who did not. However, it was shown that, even if serum inorganic fluoride (a metabolite of sevoflurane that may be involved in renal injury) concentrations exceeded 50 μmol/L during prolonged anesthesia with sevoflurane [42], no significant change in markers of renal function has been reported to date in both healthy volunteers and ICU patients [14, 29, 43, 44]. Nevertheless, clinicians should be cautious if volatile anesthetics are used for durations longer than 48 hours. Furthermore, data supporting potential beneficial pre- or post-conditioning effects of volatile anesthetics on the liver, the heart or the kidneys in the setting of anesthesia might not be transposable to ICU patients.

Most respondents evoked malignant hyperthermia as a specific contraindication to volatile anesthetics. Indeed, although its incidence is rare, malignant hyperthermia is an absolute contraindication and can be a serious and life-threatening condition. Almost half of physicians answered that they strictly avoid using volatile agents in pregnant patients. Few animal studies were published on the effects of volatile anesthetics during pregnancy and they reported conflicting on potential neurodevelopmental toxicity [45, 46]. Although their precise effects during human pregnancy are still largely unknown, an overall cautious strategy is currently recommended. Intracranial hypertension was a contraindication for 20% of respondents; although volatile anesthetics do result in cerebral vasodilatation and are therefore likely to

increase intracranial pressure, their vasodilator effect is dose-dependent and it has been shown that cerebral autoregulation remains intact during sevoflurane anesthesia in humans [47].

Previous experience of adverse events were reported by 11 (28%) respondents; none was described as life-threatening, reinforcing previous data on the safety of inhaled ICU sedation [2, 13, 38]. Severe hypercapnic acidosis due to inhaled sedation was reported by one respondent, which could be explained by the increased instrumental deadspace volume of 100 mL generated by the *AnaConDa* or *Mirus* devices [39, 40, 48], although a miniaturised version of the *AnaConDa* has been developed with an instrumental deadspace volume of 50 mL, allowing the use of a minimum tidal volume of 200 mL [37, 49]. Two respondents reported histories of headaches in some nurses, which can be compatible with exposition to environmental pollution from volatile anesthetics. Indeed, volatile anesthetics induced- pollution is a major concern for environmental protection. However, atmospheric pollution is normally avoided by the use of charcoal filters with activated carbon connected to the expiratory branch of the ventilator or of active scavenging connected to the vacuum system. Possibly due to their use of such pollution-limiting systems, only 10 respondents had concerns about the risk of air pollution and mentioned it as a reason not to use inhaled ICU sedation. Thirty-four percent of respondents stated that inhaled sedation might represent a good alternative to intravenous sedation in ICU patients; however, this rate increased to 72% when considering only the answers from current users of the technique. However, and although inhaled sedation might represent a viable alternative to some situations of shortage in intravenous sedatives, such as during the Covid-19 pandemics [50], more studies are now needed to confirm its potential benefits in ICU patients.

Our study has several limitations. First, a survey has an intrinsic bias which could only decrease with a prospective study. Indeed, such declarative surveys can only provide limited information and an observational study over a given time or repeated over time would have given more scope to clinical observations and more relevance to the real-life use of volatile anesthetics in the ICU setting. Nevertheless, the validity of the results is strengthened by the high response rate and a good distribution of the different hospital types in France among respondents. Moreover, geographical distributions of respondents and non-respondents were similar (Supplemental Content 4 of S1 File). Furthermore, this survey was designed to question ICU directors only, which limits both the non-response rate and the response bias, and may increase the validity of our findings. Second, although the oral presentation of the survey may have favored the response rate and allowed precise answers to some descriptive questions, it is possible that, for the closed questions, reported numbers might have been overestimated. Finally, these findings, which reflect the use of inhaled sedation as reported by French intensivists, may not be extrapolated to other countries with distinct ICU organizations.

## Conclusion

In conclusion, this survey highlights both good knowledge about inhaled ICU sedation and its limited use in France to date. Most respondents declared they used inhaled sedation in few patients per year and since a few years only. Differences in education and knowledge of intensivists, as well as the recent and relatively scarce literature on the use of volatile agents in the ICU, might explain the diverse practices as observed.

## Supporting information

**S1 File.**
(DOCX)

**S2 File. STROBE statement—checklist of items that should be included in reports of observational studies.**
(DOCX)

## Author Contributions

**Conceptualization:** Raiko Blondonnet, Audrey Quinson, Jean-Michel Constantin, Matthieu Jabaudon.

**Data curation:** Raiko Blondonnet, Matthieu Jabaudon.

**Formal analysis:** Céline Lambert, Bruno Pereira.

**Funding acquisition:** Raiko Blondonnet, Matthieu Jabaudon.

**Investigation:** Raiko Blondonnet, Audrey Quinson, Jean-Michel Constantin, Matthieu Jabaudon.

**Methodology:** Raiko Blondonnet, Audrey Quinson, Céline Lambert, Bruno Pereira, Jean-Michel Constantin, Matthieu Jabaudon.

**Project administration:** Raiko Blondonnet, Audrey Quinson, Matthieu Jabaudon.

**Supervision:** Raiko Blondonnet, Audrey Quinson, Jean-Michel Constantin, Matthieu Jabaudon.

**Validation:** Raiko Blondonnet, Audrey Quinson, Jean-Michel Constantin, Matthieu Jabaudon.

**Writing – original draft:** Raiko Blondonnet, Audrey Quinson, Jean-Michel Constantin, Matthieu Jabaudon.

**Writing – review & editing:** Raiko Blondonnet, Audrey Quinson, Céline Lambert, Jules Audard, Thomas Godet, Ruoyang Zhai, Bruno Pereira, Emmanuel Futier, Jean-Etienne Bazin, Jean-Michel Constantin, Matthieu Jabaudon.

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
