## [Decision Letter · Decision Letter 0]

22 Jan 2021

PONE-D-20-38460

Use of volatile agents for sedation in the intensive care unit: A national survey in France

PLOS ONE

Dear Dr. BLONDONNET,

Thank you for submitting your manuscript to PLOS ONE. After careful consideration, we feel that it has merit but does not fully meet PLOS ONE’s publication criteria as it currently stands. Therefore, we invite you to submit a revised version of the manuscript that addresses the points raised during the review process.

We look forward to receiving your revised manuscript.

Kind regards,

Corstiaan den Uil

Academic Editor

PLOS ONE

Journal Requirements:

2. Please include additional information regarding the survey or questionnaire used in the study and ensure that you have provided sufficient details that others could replicate the analyses. For instance, if you developed a questionnaire as part of this study and it is not under a copyright more restrictive than CC-BY, please include a copy, in both the original language and English, as Supporting Information.  If the original language is written in non-Latin characters, for example Amharic, Chinese, or Korean, please use a file format that ensures these characters are visible.

3. Thank you for stating the following in the Financial Disclosure of your manuscript:

"This work was supported by internal funding of the Department of Perioperative medicine,

CHU Clermont-Ferrand, France."

"The authors received no specific funding for this work"

"The authors declare that there is no conflict of interest regarding the publication of this survey. MJ is a principal investigator of the SEvoflurane for Sedation in ARds (SESAR) (ClinicalTrials.gov Identifier: NCT04235608) and the ISCA study (ClinicalTrials.gov Identifier: NCT04383730), which are co-funded and funded, respectively, by grants from Sedana Medical. JMC and MJ received fees from Sedana Medical for participation in a scientific advisory panel in 2019; MJ received consulting fees from Abbvie in 2020. Neither Sedana Medical or Abbvie has no influence in the study and collection, analysis, and interpretation of data and in writing."

Reviewers' comments:

Reviewer's Responses to Questions

**Comments to the Author**

1. Is the manuscript technically sound, and do the data support the conclusions?

Reviewer #1: Yes

Reviewer #2: Yes

Reviewer #3: Partly

2. Has the statistical analysis been performed appropriately and rigorously? 

Reviewer #1: Yes

Reviewer #2: Yes

Reviewer #3: Yes

3. Have the authors made all data underlying the findings in their manuscript fully available?

Reviewer #1: Yes

Reviewer #2: Yes

Reviewer #3: Yes

4. Is the manuscript presented in an intelligible fashion and written in standard English?

Reviewer #1: Yes

Reviewer #2: Yes

Reviewer #3: Yes

5. Review Comments to the Author

Reviewer #1: The authors performed a survey in French adult ICUs about the use and about the knowledge, perceived advantages, and disadvantages of inhaled sedation for invasively ventilated patients. The presented results are robust due to a large number of responding ICUs. The conclusions drawn are supported by the results and are of great importance and currentness in view of the ongoing Covid-19 pandemic with increased demand on artificial ventilation and sedation.

However, there are some major points that should be addressed before the paper is ready for publication:

1. Background, P4L14: As the Mirus is still quite unknown, as shown by the results of your paper, please quote at least two publications about the new device, e.g. the following

• Bomberg et al: A novel device for target-controlled administration and reflection of desflurane - the Mirus. Anaesthesia 2014, 69: 1241-50

• Bomberg et al.: AnaConDa and Mirus for intensive care sedation, SpringerPlus 2016: 5:420

• Romagnoli: The New Mirus System for Short-Term Sedation in Postsurgical ICU patients. Crit Care Med. 2017

2. Methods, 1st para: “was granted exemption from review…” - What does this mean? And how can the IRB waive the need for informed consent, if it did not know the survey?

3. Results, 1st para: How is it possible that 91% of responding ICUs are polyvalent (I would prefer the term “mixed”) and 8% were medical and none was surgical? I am not familiar with the situation of ICUs in France, but complete absence of predominantly surgical ICUs might impact the findings. Please comment on that issue.

4. Rest of results: Please write more clearly. Avoid synonyms like "participating ICUs", “surveyed ICUs” and "respondents", “users” and “units in which volatile agents were used”. Define strict terms, like “respondents” versus “non-respondents”, “users” versus “non-users”, may-be “frequent users” or “regular users” versus “occasional users.” Also, please report data always in the same way, e.g. number (percentage) or better percentage (17/45). Please check, whether more data can be displayed in figures and shorten the text.

I personally got quite confused and had to read the results section several times to understand. Please rewrite the results section.

5. Figures:

Figure 1: In the legend you write: “in surveyed ICUs in which a dedicated device was available” (do you mean: users? Or regular users?) In the figure it says: “percentage of ICU respondents” This is not the same. Also, please stick to one term: either “surveyed ICU” or “responding ICU”.

Figure 3: similar: In the legend you write “respondents who use the technique” (please use one term throughout, I suggest “users”) and in the figure “percentage of respondents.”

Figure 4: are these truly all respondents? (90% are convinced of a beneficial bronchodilatory effect but have never used inhaled sedation?)

Figure 5A: How can “respondents” be satisfied (or dissatisfied) with a technique, if they do not use it? And why are there only 40 “respondents”???

6. Discussion, P13 second half of first paragraph: it is true that there is evidence for protection by volatile anesthetics against ischemia-reperfusion injury also in the kidney. But the use of sevoflurane for (much) longer than 48 hours should be strongly discouraged, especially if fluoride levels are not monitored daily. Off label use with scientific evidence of benefit, as for isoflurane or sevoflurane up to 48 hours, is a very much different from off-label use with scientific evidence of harm.

7. Literature:

Please quote more literature from the ICU and less from anesthesia for short lasting invasive procedures, e.g. 17 (Uhlig), 31 (Tyagi), 32 (Golembiewski), 35 (Ibrahim),43 (Perbet), 38 (Ogurlu), also 37 (Pavcnik) which refers to pediatric ICU.

Inhaled sedation should not be used via face masks in critically ill patients with respiratory failure!!!

I suggest not to mention this aspect, or else to discourage this use more strongly than simply stating that it is not supported by evidence. Quoting literature from the OR, where this is done, is not helpful! The situation in the ICU is completely different (longer time, patients not fasting (!) and with respiratory failure, no close supervision by anesthesiologist).

Instead of Uhlig, Bellgardt et al., EJA 2016 33:6-13 could be used.

For cost, consumption of iso (Sackey 2004: 2.1 ml/h; Bomberg JCMC 2018, 32:639-46: 3.1 ml/h) or sevo (Röhm ICM 2008 34:1683-9: 3.2 mL/h including cost analysis!) may be quoted, as well as L’Her, which you cite.

P12 para3L5: please do not quote Chabanne as evidence for bronchodilation. We now know that the increase in work of breathing was caused by severe carbon dioxide reflection by the device in the absence of sevoflurane.

P14 para 2 line 5: “deadspace…by AnaConDa or Mirus”: please also quote:

• Bomberg et al. Volumetric and reflective device dead space of anesthetic reflectors… JCMC 2018 32:1073-80.

Only this paper evaluated AnaConDa and the Mirus and made the distinction by volumetric (internal size) and reflective (caused by CO reflection) dead space.

P14 para 2 line 5: “a miniaturised version of the AnaConDa” please quote the first descriptions of this new device:

• Bomberg et al.: Halving the volume of AnaConDa…in critically ill patients. JCMC 2018 32:639-46, and

• Bomberg et al.: Halving the volume of AnaConDa…in a test lung model Anesthesiology 2019 129:371-79.

Minor issues:

Language: please use specific terms throughout. My preference for the substances in question would be volatile anesthetics. Halogenated agents are also part of refrigerators.

Background:

p4L5: intravenous anaesthetics  intravenous sedatives

P4L9: guidelines reported  guidelines recommend

P4L17 protection of the workplace environment

Results:

P8, para2, L1-2: suggestion: “Among responding ICUs, 73% (137/1XX?) declared that they knew about the use of inhaled sedation in the ICU. Of these, 98% (135/137) knew the AnaConDa and 15% (xx/137) knew the Mirus device.”

P10Para2L6: Halogenated-related adverse events – poor style

P10 para 2 last line: “after using volatile anesthetics...", “since" may imply a causal relationship

P12 para3 L7: replace “severe” by “severely ill” or “critically ill”

Discussion:

P14para2 L10: by use of charcoal filters with activated carbon placed on the expiratory branch of the ventilator or by active scavenging connected to the vacuum system.

P14 para 2 last but one line: “inform” did you mean “confirm”?

Reviewer #2: Blondonnet and colleagues should be commended to analyze in a survey the use of inhaled ICU sedation practice in France. The topic is original. The authors reported the interest of such a method for the intensivists but only few management in the daily clinical practice . Some reasons for indications and limitation of use were reported. However some few concerns deserve comments

Main comments:

- The main drawback of this type of study is the unreliable character of the veracity/quality of the data related to the questionnaires. Physicians have a real tendency to reduce or increase the number of patients, the difficulties or ease of obtaining devices, etc. An observational study over a given time or repeated twice a year over a month, for instance, would have given more scope to clinical observations and more relevance to the indications. I think that authors should mention that point.

- It is difficult to compare the reported effects in anesthesia for the preconditioning found with halogenated and the effects sought in intensive care, in particular for ventilation, which are very far from the effects associated with anesthesia using large expiratory fraction and high MAC.It is very surprising for the reviewer to see in the table some preconditioning expected effects as reported in cardiac anesthesia for instance. The reviewer should comment about these points.

- What is the training or fellowship of the physicians responders, ICU anesthesiologists or strictly intensivists? Is there a difference in the answers between the doctors from these two different training courses?

- Did the teams that manage patients with inhaled sedation use pollution-limiting systems on the ventilator's expiratory branch?

- Did the teams that used sevoflurane and isoflurane notice a difference in the hemodynamic profile of he ICU patients or the quality of the sedation?

- Can the authors have an idea on the cost-effectiveness profile of this type of management compared to IV sedation.

Reviewer #3: The authors present results from a survey of ICUs in France on their use of inhaled sedation. They provide descriptive statistics about the use and beliefs regarding inhaled sedation from about 50% of the non-pediatric ICUs in the country. The manuscript will be strengthened if the authors consider the following points.

1. Only 50% of ICUs provided information for the survey. Do authors have any information about the non-respondents to be able to comment on the similarities and differences (besides saying the respondents are geographically distributed and so authors feel they are representative)? Do authors have reasons why certain ICUs did not participate? Was that solely due to being unable to contact the ICU director?

2. Authors provide largely descriptive results from this survey. Were there any underlying hypotheses of interest that might be tested given the information that was being asked in the survey? I ask partially because in the Discussion, the authors state "Differences in practices between medical and polyvalent ICUs...", which could be just a general statement about medical and polyvalent ICUs being different rather than specific differences identified in the survey.

3. In the presentation of results (including in the Abstract), authors present frequencies and corresponding percentages. Sometimes, these percentages are calculated out of the entire sample of respondents, sometimes out of those using inhaled sedation, sometimes out of those not using inhaled sedation. It is not always clear which subgroup is being referenced, especially when (it seems) results across different subgroups are presented in the same paragraph or section. Authors should read the results section carefully and think about the organization or clarification that can be added, to ensure that it is clear who specifically the results are about.

4. Authors should think about the importance of the various figures. In many cases, authors provide all (or most) of the information in the figure in the text as well.

Minor points:

1. 9th line under results: "whose 98% percent knew..." - since "%" is already there, authors can remove "percent". This sentence should also be rephrased as it is a bit unclear as currently written.

2. 13th line under results: "44% (17/40)", 17/40 does not equal 44%

6. PLOS authors have the option to publish the peer review history of their article (what does this mean?). If published, this will include your full peer review and any attached files.

Reviewer #1: **Yes: **Priv.-Doz. Dr. med. Andreas Meiser

Reviewer #2: No

Reviewer #3: No

---

## [Author Response · Author response to Decision Letter 0]

24 Mar 2021

RESPONSE TO EDITOR AND REVIEWER COMMENTS

Reference: MS#: PONE-D-20-38460 

Title: Use of volatile agents for sedation in the intensive care unit: A national survey in France

Raiko Blondonnet, Audrey Quinson, Céline Lambert, Jules Audard, Thomas Godet, Ruoyang Zhai, Bruno Pereira, Emmanuel Futier, Jean-Etienne Bazin, Jean-Michel Constantin, Matthieu Jabaudon

We thank the Editors and the Reviewers for their careful read and thoughtful comments on the previous manuscript version. We have carefully taken these comments into consideration in preparing our revision, and we hope the manuscript has been improved. Please find below a point-by-point response to the comments and questions.

EDITORIAL COMMENTS

Dear Dr. BLONDONNET,

Thank you for submitting your manuscript to PLOS ONE. After careful consideration, we feel that it has merit but does not fully meet PLOS ONE’s publication criteria as it currently stands. Therefore, we invite you to submit a revised version of the manuscript that addresses the points raised during the review process

REVIEWERS COMMENTS

REVIEWER #1: The authors performed a survey in French adult ICUs about the use and about the knowledge, perceived advantages, and disadvantages of inhaled sedation for invasively ventilated patients. The presented results are robust due to a large number of responding ICUs. The conclusions drawn are supported by the results and are of great importance and currentness in view of the ongoing Covid-19 pandemic with increased demand on artificial ventilation and sedation.

However, there are some major points that should be addressed before the paper is ready for publication

Major issues

C1. Background, P4L14: As the Mirus is still quite unknown, as shown by the results of your paper, please quote at least two publications about the new device, e.g. the following

Bomberg et al: A novel device for target-controlled administration and reflection of desflurane - the Mirus. Anaesthesia 2014, 69: 1241-50

Bomberg et al.: AnaConDa and Mirus for intensive care sedation, SpringerPlus 2016: 5:420

Romagnoli: The New Mirus System for Short-Term Sedation in Postsurgical ICU patients. Crit Care Med. 2017

R1. As suggested by Reviewer #1, we added these three references on the Mirus system in the revised version of the manuscript. 

C2. Methods, 1st para: “was granted exemption from review…” - What does this mean? And how can the IRB waive the need for informed consent, if it did not know the survey?

R2. According to French law and by design, such a survey did not require examination by an Ethics Committee and consequently, no informed consent was needed.

We have rephrased the sentence to avoid any misunderstanding. Thank you.

C3. Results, 1st para: How is it possible that 91% of responding ICUs are polyvalent (I would prefer the term “mixed”) and 8% were medical and none was surgical? I am not familiar with the situation of ICUs in France, but complete absence of predominantly surgical ICUs might impact the findings. Please comment on that issue.

R3. We agree that the French organization of ICU staffing can be complicated to international readers.

In France, a combined anesthesiology, critical care and perioperative medicine specialty track qualifies a MD as an “anesthesiologist and intensivist” specialist, who can work in the operating room, in the ICU, or in both.

In addition, a distinct specialty track of medical critical care trains French “medical intensivists” who can work in ICUs but cannot practice anesthesiology. Currently, ICUs managed by anesthesiologists-intensivists represent 82% of the total number of ICUs in France, that are mainly surgical and/or medical (grouped under the term “polyvalent” or “mixed”, as suggested). In contrast, ICUs managed by “medical intensivists” are usually called “medical”. Unfortunately, the proportion of anesthesiologist-intensivist staff in each participating unit was not assessed in this survey.

As suggested by Reviewer #1, we have changed the term “polyvalent” for “mixed” and added, in the Discussion paragraph, some description of ICU organization and staffing in France, in order to help the reader better understand the findings.

C4. Rest of results: Please write more clearly. Avoid synonyms like "participating ICUs", “surveyed ICUs” and "respondents", “users” and “units in which volatile agents were used”. Define strict terms, like “respondents” versus “non-respondents”, “users” versus “non-users”, may-be “frequent users” or “regular users” versus “occasional users.” Also, please report data always in the same way, e.g. number (percentage) or better percentage (17/45). Please check, whether more data can be displayed in figures and shorten the text.

I personally got quite confused and had to read the results section several times to understand. Please rewrite the results section.

R4. We thank Reviewer #1 for the comment and we apologize if our manuscript could be confusing under its previous form. 

As requested, we changed the Results section to avoid synonyms (eg, we chose the term “respondents” as suggested) and we rephrased all numbers using percentages (x/x) to improve readability.

We also significantly re-organized the Results section to avoid redundancy and illustrate more data with figures, while reducing the text.

C5. Figures:

Figure 1: In the legend you write: “in surveyed ICUs in which a dedicated device was available” (do you mean: users? Or regular users?) In the figure it says: “percentage of ICU respondents” This is not the same. Also, please stick to one term: either “surveyed ICU” or “responding ICU”.

Figure 3: similar: In the legend you write “respondents who use the technique” (please use one term throughout, I suggest “users”) and in the figure “percentage of respondents.”

Figure 4: are these truly all respondents? (90% are convinced of a beneficial bronchodilatory effect but have never used inhaled sedation?)

Figure 5A: How can “respondents” be satisfied (or dissatisfied) with a technique, if they do not use it? And why are there only 40 “respondents”???

C5. Figure 1: As previously answered, and to be in accordance with the body of the manuscript (where the word “respondents” was used), we changed the legend with the term “responding intensive care units”. Figure 1 showed the number of patients in whom inhaled sedation was used each year but only in responding ICU in which a dedicated device was available (i.e., either AnaConDa or Mirus). “n=40” reported the total number of responding ICU with such a device was available. We deleted “n=40” in the legend to avoid any confusion with the number of patients receiving inhaled sedation.

Figure 3: As suggested, we have added “percentage of respondents” to the figure and changed the legend to add information on “users”.

Figure 4: Figure 4 reported data from respondents who declared they were aware of potential benefits of inhaled ICU sedation (n=129). In this sample of respondents, 88% (114/129) reported bronchodilation as a potential benefit; this percentage was very close between respondents who had a device available in their unit (90%, 35/39) and those who did not (88%, 79/90) . We have rephrased the revised manuscript to better detail these results.

Figure 5: We thank Reviewer #1 for this suggestion. We added a figure (new Figure 5), and previous Figure 5 is now Figure 6 in the revised manuscript. Figure 6A only reported satisfaction of respondents who had an inhaled ICU sedation device available and, therefore, used inhaled sedation in practice, which explains “n=40”. We rephrased the legend of Figure 6 to improve the understanding by the readers.

C6. Discussion, P13 second half of first paragraph: it is true that there is evidence for protection by volatile anesthetics against ischemia-reperfusion injury also in the kidney. But the use of sevoflurane for (much) longer than 48 hours should be strongly discouraged, especially if fluoride levels are not monitored daily. Off-label use with scientific evidence of benefit, as for isoflurane or sevoflurane up to 48 hours, is very much different from off-label use with scientific evidence of harm.

R6. We thank Reviewer #1 for the comment. As suggested, we rephrased this paragraph to highlight that clinicians should be very cautious if they use sevoflurane or isoflurane for ICU sedation for more than 48 hours.

C7. Literature:

Please quote more literature from the ICU and less from anesthesia for short lasting invasive procedures, e.g. 17 (Uhlig), 31 (Tyagi), 32 (Golembiewski), 35 (Ibrahim),43 (Perbet), 38 (Ogurlu), also 37 (Pavcnik) which refers to pediatric ICU.

Inhaled sedation should not be used via face masks in critically ill patients with respiratory failure!!!

I suggest not to mention this aspect, or else to discourage this use more strongly than simply stating that it is not supported by evidence. Quoting literature from the OR, where this is done, is not helpful! The situation in the ICU is completely different (longer time, patients not fasting (!) and with respiratory failure, no close supervision by anesthesiologists).

Instead of Uhlig, Bellgardt et al., EJA 2016 33:6-13 could be used.

For cost, consumption of iso (Sackey 2004: 2.1 ml/h; Bomberg JCMC 2018, 32:639-46: 3.1 ml/h) or sevo (Röhm ICM 2008 34:1683-9: 3.2 mL/h including cost analysis!) may be quoted, as well as L’Her, which you cite.

R7. As suggested by Reviewer #1, we edited the references and removed articles from anesthesia to increase data from ICU using the reference of Bellgardt. We also used all references suggested by Reviewer #1 regarding the cost of inhaled ICU sedation. Thank you.

We fully agree with Reviewer #1 that inhaled sedation should not be used via face mask in critically ill patients with respiratory failure. As suggested, we removed this part to avoid any misunderstanding and to strongly discourage this use. 

C9. P12 para3L5: please do not quote Chabanne as evidence for bronchodilation. We now know that the increase in work of breathing was caused by severe carbon dioxide reflection by the device in the absence of sevoflurane.

R9. As suggested by Reviewer #1, we rephrased the related paragraph highlighting that the increase in work of breathing was caused by severe CO2 reflection by the device in the absence of sevoflurane and not by bronchodilation. We also quoted the study of Sturesson et al. (Br J Anaesth 2014) to reinforce this point. 

C10. P14 para 2 line 5: “deadspace…by AnaConDa or Mirus”: please also quote:

Bomberg et al. Volumetric and reflective device dead space of anesthetic reflectors… JCMC 2018 32:1073-80. Only this paper evaluated AnaConDa and the Mirus and made the distinction by volumetric (internal size) and reflective (caused by CO reflection) dead space.

R10. As suggested by Reviewer #1, we added this reference to the revised manuscript.

C11. P14 para 2 line 5: “a miniaturised version of the AnaConDa” please quote the first descriptions of this new device:

Bomberg et al.: Halving the volume of AnaConDa…in critically ill patients. JCMC 2018 32:639-46, and

Bomberg et al.: Halving the volume of AnaConDa…in a test lung model Anesthesiology 2019 129:371-79.

R11. As suggested by Reviewer #1, we added these references to the revised manuscript.

Minor issues:

C12. Language: please use specific terms throughout. My preference for the substances in question would be volatile anesthetics. Halogenated agents are also part of refrigerators.

R12. As suggested by Reviewer #1, we used the term “volatile anesthetics” rather than “halogenated agents” in our revision.

C13. Background:

p4L5: intravenous anaesthetics  intravenous sedatives

P4L9: guidelines reported  guidelines recommend

P4L17 protection of the workplace environment

R13. As suggested by Reviewer #1, we changed these sentences in the Background section.

C14. Results:

P8, para2, L1-2: suggestion: “Among responding ICUs, 73% (137/1XX?) declared that they knew about the use of inhaled sedation in the ICU. Of these, 98% (135/137) knew the AnaConDa and 15% (xx/137) knew the Mirus device.”

R14. As suggested by Reviewer #1, we have rephrased this paragraph in the Results section.

C15. P10Para2L6: Halogenated-related adverse events – poor style

P10 para 2 last line: “after using volatile anesthetics...", “since" may imply a causal relationship

P12 para3 L7: replace “severe” by “severely ill” or “critically ill”

R15. As suggested by Reviewer #1, these sentences have been changed.

C16. Discussion:

P14para2 L10: by use of charcoal filters with activated carbon placed on the expiratory branch of the ventilator or by active scavenging connected to the vacuum system.

P14 para 2 last but one line: “inform” did you mean “confirm”?

R16. As recommended by Reviewer #1, the sentences have been changed in the Discussion section. Thank you.

REVIEWER #2: Blondonnet and colleagues should be commended to analyze in a survey the use of inhaled ICU sedation practice in France. The topic is original. The authors reported the interest of such a method for the intensivists but only few management in the daily clinical practice . Some reasons for indications and limitation of use were reported. However some few concerns deserve comments

Main comments:

C17. The main drawback of this type of study is the unreliable character of the veracity/quality of the data related to the questionnaires. Physicians have a real tendency to reduce or increase the number of patients, the difficulties or ease of obtaining devices, etc. An observational study over a given time or repeated twice a year over a month, for instance, would have given more scope to clinical observations and more relevance to the indications. I think that authors should mention that point.

R17. We thank Reviewer #2 and fully agree with this comment. Actually, in order to better and more accurately describe the use of inhaled ICU sedation, our group is currently completing an international observational study (ClinicalTrials.gov Identifier: NCT04383730), but in the specific setting of COVID-19-related acute respiratory distress syndrome. 

As suggested, we added this limitation of our survey to the Discussion section.

C18. It is difficult to compare the reported effects in anesthesia for the preconditioning found with halogenated and the effects sought in intensive care, in particular for ventilation, which are very far from the effects associated with anesthesia using large expiratory fraction and high MAC.It is very surprising for the reviewer to see in the table some preconditioning expected effects as reported in cardiac anesthesia for instance. The reviewer should comment about these points.

R18. We thank Reviewer #2 for the comment. We added a comment to these specific points in the Discussion (page 15 of the revised manuscript).

C19. What is the training or fellowship of the physicians responders, ICU anesthesiologists or strictly intensivists? Is there a difference in the answers between the doctors from these two different training courses?

R19. As previously answered to Reviewer #1 in C3, the French organization of ICU staffing can be complicated to international readers. 

In France, a combined anesthesiology, critical care and perioperative medicine specialty track qualifies a MD as an “anesthesiologist and intensivist” specialist, who can work in the operating room, in the ICU, or in both.

In addition, a distinct specialty track of medical critical care trains French “medical intensivists” who can work in ICUs but cannot practice anesthesiology. Currently, ICUs managed by anesthesiologists-intensivists represent 82% of the total number of ICUs in France, that are mainly surgical and/or medical (grouped under the term “polyvalent” or “mixed”, as suggested). In contrast, ICUs managed by “medical intensivists” are usually called “medical”. Unfortunately, the proportion of anesthesiologist-intensivist staff in each participating unit was not assessed in this survey.

In order to better characterize the population of respondents and their levels of training, we reported in the Results section whether respondents had received specific training on inhaled sedation, either originated from the companies developing the AnaConda or the Mirus devices and/or through scientific conferences. 

As suggested by Reviewer #2, we also discussed these points in the revised Discussion. 

C20. Did the teams that manage patients with inhaled sedation use pollution-limiting systems on the ventilator's expiratory branch?

R20. We thank Reviewer #2 for this important comment and we agree that volatile anesthetics induced-pollution is a major concern for environmental protection. Unfortunately, we did not assess this specific point in our survey, as we postulated that the use of a pollution-limiting system (scavenger) connected to the ventilator’s expiratory branch was part of good clinical practice, as recommended by manufacturers of inhaled ICU devices.

Indeed, and possibly due to their use of such pollution-limiting systems, only 10/147 respondents on the survey had concerns about the risk of air pollution, which could be felt as a reason not to use inhaled ICU sedation.

We have discussed this important point in the revised Discussion (page 16).

C21. Did the teams that used sevoflurane and isoflurane notice a difference in the hemodynamic profile of the ICU patients or the quality of the sedation?

R21. We thank Reviewer #2 for this interesting comment; unfortunately, we did not ask specific questions on the hemodynamic profiles or sedation efficacy in our survey.

C22. Can the authors have an idea on the cost-effectiveness profile of this type of management compared to IV sedation.

R22. We thank Reviewer #2 for this comment. 

One fifth of the respondents answered that they did not use inhaled sedation because of its related cost. 

Even if inhaled ICU sedation might be perceived as “more costly” at present, reductions in indirect costs through shortened times to liberation from mechanical ventilation, patient recovery, and ICU discharge following the judicious use of volatile anesthetics, combined with a reduction in the need for other treatments (such as opioid use), are plausible but often difficult to demonstrate in studies.

Regarding direct costs, in some patients in whom sedation is deemed more difficult, inhaled sedation may significantly decrease costs of sedation, with daily costs being similar between midazolam or isoflurane, and the cost difference could favor inhaled sedation in the subgroup of patients who required high doses of midazolam (L’Her et al.). We further discussed this point in the Discussion section (page 13).

REVIEWER #3: The authors present results from a survey of ICUs in France on their use of inhaled sedation. They provide descriptive statistics about the use and beliefs regarding inhaled sedation from about 50% of the non-pediatric ICUs in the country. The manuscript will be strengthened if the authors consider the following points.

Major points

C23. Only 50% of ICUs provided information for the survey. Do authors have any information about the non-respondents to be able to comment on the similarities and differences (besides saying the respondents are geographically distributed and so authors feel they are representative)? Do authors have reasons why certain ICUs did not participate? Was that solely due to being unable to contact the ICU director?

R23. We thank Reviewer #3 for this valuable comment, and we regret we did not assess this detailed information for non-respondent centers.

Indeed, after a first contact by telephone, and upon request from the ICU Medical Chief office, some questionnaires were emailed individually. In the absence of a reply after one month, a second contact with the ICU Director was performed by telephone and followed by a third call when necessary.

Among the units that were contacted, some centers did not want to participate in the survey and/or did not have time to respond. Furthermore, when no answer was received after the third reminder by phone, the center was considered as “non-respondent”. Unfortunately, we did not distinguish centers which did not want to participate from those that did not respond to our three invitations. The only element that was collected was the geographical distribution of non-respondents, which has been added as a limitation and a supplemental content (Supplemental Content 4) to our revised manuscript.

C24. Authors provide largely descriptive results from this survey. Were there any underlying hypotheses of interest that might be tested given the information that was being asked in the survey? I ask partially because in the Discussion, the authors state "Differences in practices between medical and polyvalent ICUs...", which could be just a general statement about medical and polyvalent ICUs being different rather than specific differences identified in the survey.

R24. We agree with Reviewer #3 and, as previously answered to both Reviewer #1 and Reviewer #2, we agree the French organization of ICU staffing can be complicated to international readers; additional information has been added to the Discussion section to decrease the risk of any misunderstanding. The paragraph related to this statement about “medical” and “polyvalent” ICUs has, therefore, been rephrased.

C25. In the presentation of results (including in the Abstract), authors present frequencies and corresponding percentages. Sometimes, these percentages are calculated out of the entire sample of respondents, sometimes out of those using inhaled sedation, sometimes out of those not using inhaled sedation. It is not always clear which subgroup is being referenced, especially when (it seems) results across different subgroups are presented in the same paragraph or section. Authors should read the results section carefully and think about the organization or clarification that can be added, to ensure that it is clear who specifically the results are about.

R25. We thank Reviewer #3 and apologize if our previous presentations of results could be confusing. As also suggested by Reviewer #1 and Reviewer #2, we carefully reorganized and rephrased the Results section to improve readability.

C26. Authors should think about the importance of the various figures. In many cases, authors provide all (or most) of the information in the figure in the text as well.

R26. As suggested by Reviewer #3, we rephrased the Results section to avoid redundancies between the text and the figures.

Minor points:

C27. 9th line under results: "whose 98% percent knew..." - since "%" is already there, authors can remove "percent". This sentence should also be rephrased as it is a bit unclear as currently written.

R27. We apologize for this mistake and, as previously suggested by Reviewer #1, the sentence has been rephrased. Thank you. 

C28. 13th line under results: "44% (17/40)", 17/40 does not equal 44%

R28. Thank you for pointing out this mistake. We apologize and have corrected the percentage that was wrong in our previous version.

---

## [Editor Report · Decision Letter 1]

29 Mar 2021

Use of volatile agents for sedation in the intensive care unit: A national survey in France

PONE-D-20-38460R1

Dear Dr. BLONDONNET,

We’re pleased to inform you that your manuscript has been judged scientifically suitable for publication and will be formally accepted for publication once it meets all outstanding technical requirements.

Kind regards,

Corstiaan den Uil

Academic Editor

PLOS ONE
---

## [Editor Report · Acceptance letter]

5 Apr 2021

PONE-D-20-38460R1 

Use of volatile agents for sedation in the intensive care unit: A national survey in France 

Dear Dr. Blondonnet:

I'm pleased to inform you that your manuscript has been deemed suitable for publication in PLOS ONE. Congratulations! Your manuscript is now with our production department. 

Kind regards, 

on behalf of

Dr. Corstiaan den Uil 

Academic Editor

PLOS ONE